# Identification of elite rice lines with better breeding values using genomic prediction and multi-trait genotype ideotype distance index (MGIDI) for grain yield under irrigation cropping system

**S. S. Chaity**[1], **M. R. Islam**[2], **M. Faruquee**[2], **J. U. Ahmed**[3], **A. K. M. Aminul Islam**[1*]

**1** Department of Genetics and Plant Breeding, Gazipur Agricultural University, Gazipur, Bangladesh, **2** International Rice Research Institute (IRRI), Dhaka, Bangladesh, **3** Department of Crop Botany, Gazipur Agricultural University, Gazipur, Bangladesh

* aminulgpb@gau.edu.bd

## Abstract

For a rice breeding program to produce the desired genotypes, parental selection is the most important factor. To select elite parental materials, this study evaluated 200 International Rice Research Institute (IRRI) developed advanced breeding lines during 2022 Wet Season (WS-T. Aman) and 2023 Dry Season (DS-Boro) along with two sets of check (10 global and 6 different local for each season) using an alpha lattice design with two replications in the research field of Gazipur Agricultural University, Gazipur-1706. Among the breeding lines, significant genetic variation was found and according to heritability and genetic advance analysis, additive gene action controls plant height (PH), panicle length (PL), spikelet fertility (SF), thousand grain weight (TGW), and grain yield (t/ha) (GYTH). Correlation coefficients revealed that in both season yield has a positive relationship with panicle length (0.20**, 0.35***), spikelet fertility (0.19**, 0.23***), days to maturity (DTM) (0.37***, 0.24***), and days to 50% flowering (DFF) (0.34***, 0.29***). In this study, Principal Component Analysis (PCA) revealed 4 & 3 PCs in 2022WS and 2023DS contributing to 66.90% & 61.90% of total variability with eigenvalues>1. The Multi-Trait Genotype–Ideotype Distance Index (MGIDI) analysis results a total genetic gain of 28.51% & 25.86% in 2022WS & 2023DS and identified IR19A8066, IR19A8052, IR19A7440, IR19A9061, and IR19A9054 genotypes as valuable resources for developing recombinant populations, aligning with sustainable and effective crop improvement strategies. For parental selection, IR19A8054, IR19A8052, IR19A7501, IR19A8047, IR19A8066, IR19A7531, IR19A7523, IR19A7541, IR19A7510 with Genomic Estimated Breeding Value (GEBV)>0.50 for yield are preferred. The maximum yield (5.34 t/ha) in 2022WS was generated by check RABI dhan1, while 5.12 t/ha was produced by IR19A7664, and more than 4.50 t/ha was produced by IR19A9054, IR19A7408, and IR19A7440.

**Data availability statement:** All relevant data are within the paper and its Supporting Information files.

**Funding:** The author(s) received no specific funding for this work.

**Competing interests:** Author(s) declares that there is no potential conflict of interest.

When compared to check BRRI dhan89 (7.97 t/ha), IR19A7339 delivered the highest in 2023DS (8.67 t/ha), followed by IR19A7510, IR19A7523, IR19A8047, and IR19A8066 (>8 t/ha). With positive GEBVs of 0.19, 0.32, and 0.62, respectively, IR19A9054, IR19A9212, and IR19A8066 demonstrated superior yield in both rice growing seasons.

## Introduction

Rice is one of the major cereal grains farmed as a staple diet for more than half of the world's population. It is a semi-aquatic grass plant that belongs to the Gramineae (Poaceae) family's genus *Oryza* [1,2]. Due to its prominence as one of the primary centres for rice cultivation, South Asia has been referred to as the "food bowl" and "basket" of Asia [3]. In Southeast Asia, about 465 thousand metric tons of milled rice have been produced overall. After China and India, who produce 151 and 146 thousand metric tons of rice, respectively, Bangladesh has achieved the third place in the world for rice production with an output of 37.50 thousand metric tons [4]. Rice production in the country can be raised to 46.90 MT in 2030, 54.09 MT in 2040 and 60.85 MT in 2050 with combined contributions of yield improvements by enhanced varietal potential, reduction in existing yield gap and production increase [5]. More than 70% of Bangladesh's rural population and the country's economic structure are reliant on rice as the primary crop. But according to the BBS [6], the population is increasing by about 1.20% annually. By 2050, the global population estimated will reach to 9 billion, assuming current pace of growth continues [7]. As a result, researchers projected that rice production will drop by 12–14% by 2050, which necessitate to alleviate the problems of climate change and safeguard the food security [8]. To fulfil the demand for food from the expanding population, rice production must thus be increased. Rice productivity improvement is one of the main pillars of food safety, particularly for Asia and Africa [9]. It is a good source of calorie (one fifth) and protein (15%) [10,11]. About 164.32 million of peoples are directly or indirectly consumed rice [5]. The current rice consumption rate is 148 kg person$^{-1}$ year$^{-1}$ which is supposed to be decreased to 133 kg person$^{-1}$ year$^{-1}$ by the year 2040. In the course of time, population of Bangladesh will reach 215.4 million and then 44.60 M. tons of rice will be required to feed the people [12]. Consequently, the breeding strategy for quantitative traits must employ principles of quantitative genetics to achieve an annual 1.50% increase in grain production, in line with the current rising rate of rice consumption [13].

The purpose of plant breeding is to assemble more desirable combinations of genes or traits in new varieties, and parents should have performed as magnificent contributors to one or more traits that are being targeted in the breeding program [14,15]. Genetic gain is a crucial metric to assess the effectiveness of the breeding effort and track its progress. The rate of genetic gain attained by the breeding program will be extremely helpful in directing future breeding tactics, allocating resources, and accelerating the production of varieties for increased genetic gains [16]. The major targeted attributes for choosing parental genotypes are higher performance, wide adaptability, and yield stability [17]. Traditional breeding methods

require 10–12 years for obtaining superior varieties [18]. Higher rates of genetic gains can only be achieved by a comprehensive and methodical breeding effort that incorporates contemporary instruments and technologies. It is imperative to implement a population improvement breeding plan based on an elite x elite parent scheme that incorporates cutting-edge technologies such as rapid recycling, high-throughput phenotyping, and genomic selection (GS) [16,19].

The rate at which the genetic improvement occurs is often referred to as genetic gain and in order to deliver improved varieties to the farmers of the twenty-first century, the rate of genetic gain in rice must accelerate relative to twentieth century levels [8]. Genomic selection (GS), a substitute means of marker assisted selection (MAS) having the capacity to increase genetic gain, reduce the breeding time and thus, its efficiency is much higher. It is used to predict genomic estimated breeding values (GEBVs) for rice based on all markers on the genome are used [20]. Meuwissen *et al.* (2001) [21] reported that it is emerged to increase prediction accuracy of traits controlled by polygene using marker information. The main aim of GS is to increase the ability to predict traits that represents a strong relationship between the observed and predicted phenotype [20]. It increases the selection accuracy of the individuals by creating GEBVs based on marker information [22]. The breeder uses the GEBV values in conjunction with trait marker data to choose lines for advanced testing and parents for the following breeding cycle [23]. The required genetic gain rates of 1.5% or higher are far higher than the existing rate of genetic gains seen in the salinity breeding effort [19]. By 2050, rice's rate of genetic gain will have increased to 2.50% or more [16]. By concentrating the molecular breeding strategy on well-known high-value haplotypes and employing a genomics-enabled rapid recurrent selection strategy to improve quantitative traits primarily through accelerated breeding cycles, more recent approaches seek to incorporate the concepts of quantitative genetics into the breeding strategy [13].

Breeders confront challenges in the realm of crop improvement, specifically focusing on the intricate task of designing an ideotype—a genotype amalgamating diverse attributes for optimal performance [24]. Genetic gain is central to plant breeding, guiding the direction of crop improvement programs. Relying on a limited number of traits for selection often overlooks gains in other important characteristics, prompting breeders to aim for ideotypes—genotypes that integrate multiple desirable traits for optimal performance [25,26]. Although various selection indices, such as the widely used Smith–Hazel (SH) index, have been developed to aid multi-trait selection by leveraging phenotypic and genotypic covariance matrices along with economic weights [27,28], the SH index faces critical limitations. These include multi-collinearity—common with highly correlated traits—which results in ill-conditioned matrices and biased coefficient estimates, ultimately compromising the accuracy of genetic gain predictions [29,30]. Additionally, assigning realistic economic weightings to traits remains a complex challenge [31]. In response, Olivoto and Nardino (2021) [26] introduced the multi-trait genotype ideotype distance index (MGIDI), a robust alternative that addresses multicollinearity by evaluating all traits simultaneously and measuring genotype–ideotype distances. MGIDI aligns with ideotype breeding principles and has shown promise in enhancing genetic gains by identifying superior genotypes based on a comprehensive trait evaluation. The MGIDI model-based analysis has been used in rice [2,24,32–34], wheat [35–38], barley [39], maize [40,41], sesame [42], soybean [43], chickpea [44], cucumber [45], and other crops as well. This study aims to utilize the MGIDI index to identify rice accessions with high yield and marking a promising advancement in the field of multivariate selection indices by evaluating the strengths and weakness of the tested genotype for future breeding efforts.

The present investigation was undertaken with a view to study the extent of variation exists among the genotypes. Precise objectives are outlined- (i) to select higher breeding value genotypes as parental materials for further improvement of the performance of new breeding population by increasing the genetic gain; and (ii) to identify potential genotypes for yield and traits associated with yield simultaneously using diverse multivariate approaches.

## Materials and methods

### Plant materials

A set of 200 advanced breeding lines of $F_7$ generation developed at IRRI (International Rice Research Institute, Philippines) along with 10 global checks with season wise two different sets of popular rice cultivars as check varieties, were

used as experimental materials. Each set of checks contain 6 different cultivars for each season (2022WS- BRRI dhan49, BRRI dhan75, BRRI dhan87, Binadhan-11, Binadhan-17, Rabi Dhan-1 and 2023DS- BRRI dhan28, BRRI dhan29, BRRI dhan67, BRRI dhan88, BRRI dhan89, BRRI dhan92) which were selected accordingly with the rice growing season. (S1 Table)

## Experimental site and design

The present research was conducted at the experimental field of the Department of Genetics and Plant Breeding, Gazipur Agricultural University (GAU), Gazipur, during 2022WS (wet season – T. Aman) and 2023DS (dry season – Boro) using alpha lattice experimental design with two replications. The experimental field was divided into six blocks, subdivided into 72 plots per block, and a total of 432-unit plots were made. Each genotype was transplanted in 4 rows maintaining a spacing of 20 cm x 20 cm. The length and width of each unit plot were 5.40 m and 0.80 m. A space of 0.60 m was kept between two adjacent blocks. Healthy seedlings of 24 days old in 2022WS and 30 days old 2023DS seasons were transplanted in separate plots in the experimental field.

## Climate and soil management

The experimental site is in a subtropical climate region, characterized by heavy rainfall from July to September and scanty during the rest of the year with a gradual fall in temperature from September. The record of air temperature (°C), relative humidity (RH %), sun-shine (hr), and rainfall (mm) during the period of the experiment was collected from Bangladesh Rice Research Institute, Gazipur and graphically shown here the weather variation in both seasons (Fig 1).

The soil is the Shallow Red Brown Terrace type under the Salna series of Madhupur Tract [46], which belongs to the Agro Ecological Zone (AEZ) 28. The soil is characterized by silty clay with pH value of 6.5, containing a lot of aluminium and iron. It has less amount of organic matter, nitrogen, phosphorus, and lime. The Zn and Fe content of the soil is 0.12% and 0.34% [47]. Adequate soil fertility was ensured by applying manures and fertilizer in the recommended dose [48].

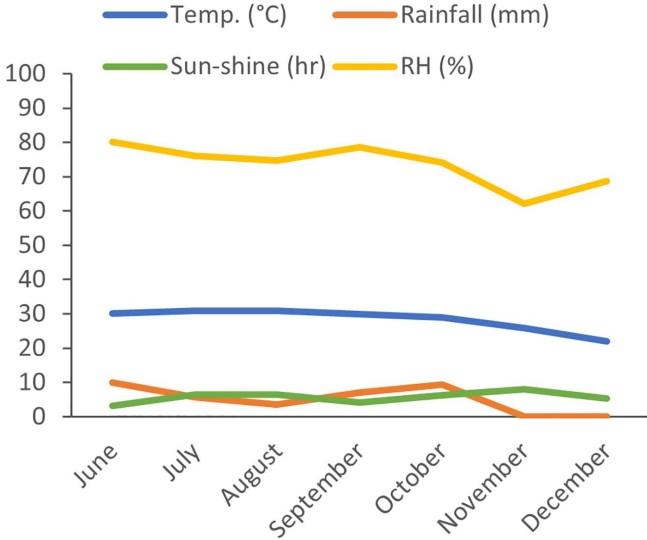

**Fig 1. Monthly average weather data of 2022WS (T. Aman) and 2023DS (Boro).**

## Data collection

A total of nine yield and yield contributing traits viz. days to 50% flowering (DFF), days to maturity (DTM), plant height (PH), effective tillers per hill (ETH), panicle length (PL), spikelet fertility (SF), grain length/width ratio (LWR), thousand grains weight (TGW), grain yield (t/ha) (GYTH) was measured based on Standard Evaluation System (SES) for rice. Five plants were randomly selected from each plot excluding border rows from both sides. Grain yield was adjusted at 14% moisture content.

## Statistical analysis

**Basic analysis.** Every form of statistical analysis was performed using R-4.0.3 for win (http://CRAN.R-project. org/) (accessed on January 11, 2023) in Rstudio-1.3.1093 (https://rstudio.com/) (accessed on January 11, 2023). Statistical significance was determined at a p-value threshold of less than 0.001. The graphical representation via boxplots summarizes multiple variable distributions, aiding in detecting relationships, comparing central tendency and variability, and aiding in exploratory data analysis and hypothesis testing [49]. The R packages 'ggplot2', 'ggpubr', and 'reshape2' were used to create boxplot.

The genotypic and phenotypic variance, genotypic coefficient of variation (GCV), phenotypic coefficient of variation (PCV), heritability in broad sense ($h^2b$), genetic advance (GA), genetic advance in percentage of mean (GAPM) were estimated following Singh and Chaudhary (1977) [50]. The estimates of GCV and PCV were classified as low, medium and high discussed by Sivasubramanian and Madhavamenon (1973) [51]. Heritability in broad sense was classified following Elrod and Stanfield (2002) [52] whereas genetic advance was classified according to methods given by Johnson *et al.* (1955) [53].

Correlation analysis not only helps identify patterns and dependencies in datasets by quantifying linear relationships between variables but also provides a holistic view of interdependencies and aids in variable selection [54]. The Pearson's correlation coefficients were computed separately for both growing conditions. The R package 'metan' was employed to visualize the correlation using the function corr_coef [55].

Principal component analysis (PCA), a versatile tool of multivariate analysis, offers insights into the structure of complex datasets not only by reducing the dimensionality of the datasets while remaining most of the variability but also identifies the key components driving the variation [56]. The R packages 'ggplot2', 'grid Extra', 'factoextra', 'ggbiplot' and 'corrplot' were used to extract the Eigen value and to visualize the PCA variable plot. The principal components (PCs) with eigenvalues >1 was considered as significant and the contribution (%) of all traits in those significant PCs was evaluated to select the key traits for succeeding multivariate approaches. Genetic diversity was determined by applying K-means clustering method.

## MGIDI analysis

Statistical analysis for the Multi-Trait Genotype–Ideotype Distance Index (MGIDI) utilized the R Package 'metan' version 1.18.0, developed by Olivoto and Lúcio [55]. The analysis was carried out in R version 4.3.1 where MGIDI calculating main steps are as follows:

## Rescaling traits

• Adjust each trait's scale so that they all range from 0 to 100. This ensures a uniform comparison and interpretation of trait values.

## Ideotype planning

• Define an ideotype by specifying target values for traits based on known or desired characteristics. This step involves determining the ideal combination of trait values for a genotype.

## Computing Genotype Distance to Ideotype

• Calculate the distance between each genotype and the planned ideotype. This distance measurement quantifies how closely a genotype aligns with the desired trait values set in the ideotype.

$$MGIDI_i = \left[ \sum_{j=1}^{f} (\gamma_{ij} - \gamma_j)^2 \right]^{0.5}$$

Where, MGIDI i is the distance index for the $i^{th}$ genotype; $\gamma_{ij}$ Score of the $i^{th}$ genotype in the $j^{th}$ factor (where i = 1, 2, 3, … g; j = 1, 2, …, f). $\gamma_j$ is the $j^{th}$ score of the ideotype.

Proportion of the MGIDI index of the $i^{th}$ genotype explained by the $j^{th}$ factor ($\omega_{ij}$) is used to show the strength and weakness of genotypes.

$$\omega_{ij} = \frac{\sqrt{Dij2}}{\sum_{j=1}^{f} \sqrt{Dij2}}$$

Where, $D_{ij}$ is the distance between the $i^{th}$ genotype and the ideotype for the $j^{th}$ factor. Low contributions of a factor indicate that the traits within such a factor are close to the ideotype.

## Breeding value estimation

GS involves identifying and choosing individuals with better breeding values using prediction models created by establishing a correlation between genotype and phenotype in a breeding population of interest. Breeding values are derived from Best Linear Unbiased Predictors (BLUPs) as the sum of BLUPs for all markers. So, it could be presented by the following way:

$$\text{Breeding value} = \text{BLUP} + \text{Marker effects} / \text{pedigree relationship}$$

In GS, a model predicting the breeding values of a target trait based on genome-wide marker genotypes is used for the selection. Now, let $x_i$ and $y_i$ represent the genome-wide marker genotype and breeding values of individual i, respectively. Then, a prediction model $f(x_i; \theta)$ can be described as:

$$y_i = f(x_i; \theta) + e_i$$

where $\theta$ is a vector of parameters included in the model and $e_i$ is a residual. Single nucleotide polymorphisms (SNPs) are generally used as genome-wide markers described by Juma et al. [13], Whole-genome genotyping for the 200 advanced rice lines was carried out by a 1k-RiCA panel (SNPs marker) test using Genotyping By Sequencing (GBS) technology described by Elshire *et al.*[57] using 995 SNP markers. Genotyping was done at the Agriplex genomic, Cedar Avenue, Suite 250, Cleveland, 011444106, USA.

## Results

### Mean performance and ANOVA

The means of the 216 genotypes for nine yield attributing traits are shown in S2 Table and represented by a box plot (Fig 2).

The box plot descriptions of the observational yield trial (OYT) of the 216 advanced rice breeding lines revealed identical in both seasons (Fig 2). The box plots generated for all the 9 characters in OYT exhibited a slight deviation from the

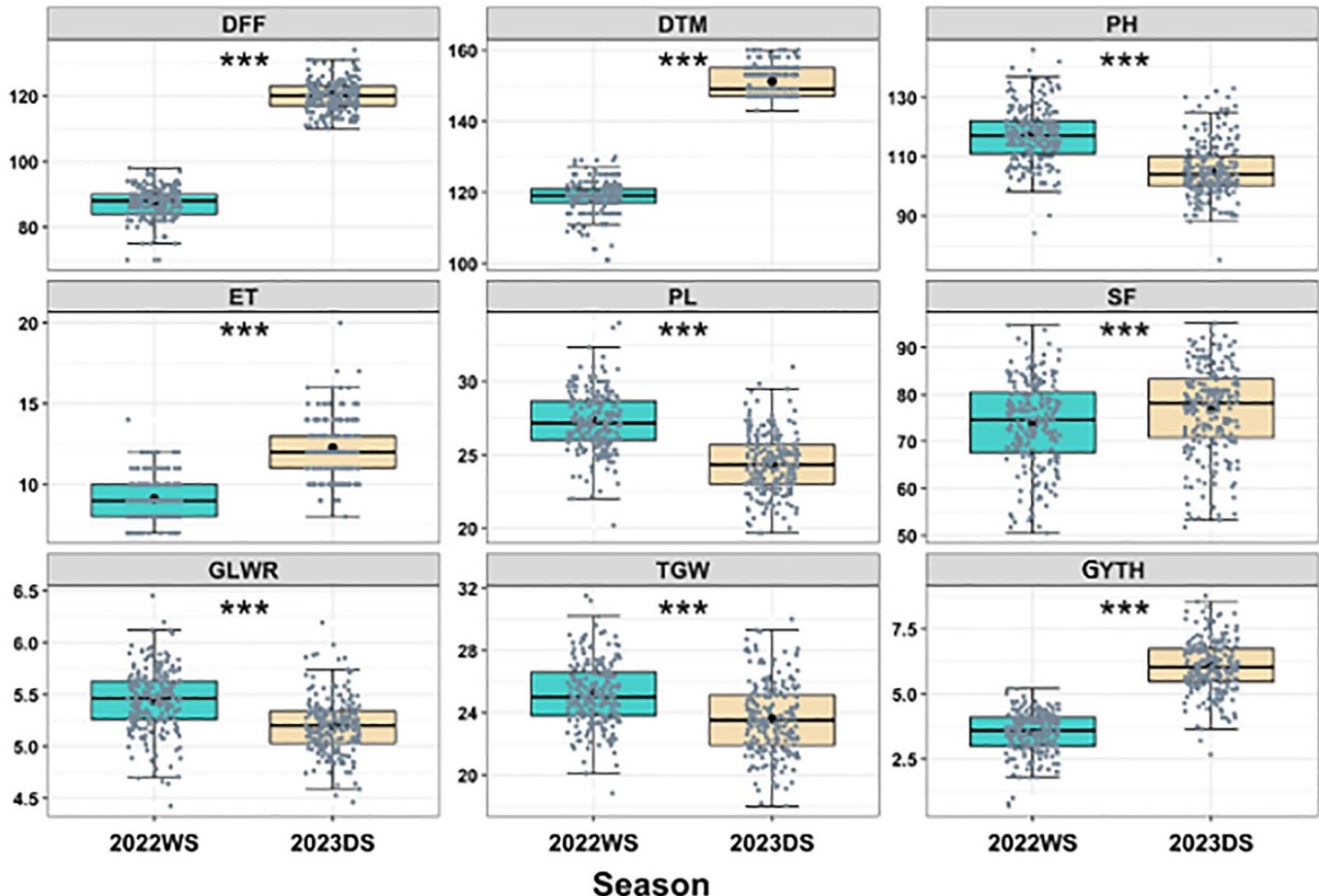

**Fig 2. Boxplot displaying quartile distribution of nine yield attributing traits in both seasons.** DFF- Days to 50% flowering; DTM- Days to Maturity; PH- Plant Height (cm); ET- Effective Tillers per Hill; PL- Panicle Length (cm); SF- Spikelet's Fertility (%); LWR- Grain Length Width Ratio; TGW- Thousand Grain Weight (g); GYTH- Grain Yield (t/ha); WS- Wet Season; DS- Dry Season. *** represents 0.001 significant level.

normal distribution. Almost all studied trait boxes were narrow compared to other traits. It was also observed in both season that all characters were skewed towards the minimum score except SF. The median was slightly skewed towards the positive side for the character, e.g., DFF and GYTH, and highly skewed for the character, e.g., DTM. It was also observed that all the characters exhibited outliers.

The Analysis of variance (ANOVA) results revealed that the mean sum of squares due to studied accessions differed significantly for all the characters under study (S2 Table). In 2022WS, the highest yield was observed in local check RABI dhan1 (5.34 t/ha) followed by IR19A7664 (5.12 t/ha), and the lowest yield was found in global check IRRI 168 (0.74 t/ha) and IR19A7428 (1.22 t/ha). The yield was affected by a tropical cyclone namely 'Sitrang' that hit at the ripening stage resulting in lodging of plants as well as shattering of grains. In 2023DS, the highest yield was observed in genotype IR19A7339 (8.67 t/ha), and the lowest yield was noticed in the variety IR19A7437 (2.50 t/ha).

## Genetic variability

For majority of the traits, the current study reveals that phenotypic coefficient variance (PCV) was greater (Table 1) but near to their matching genotypic coefficient of variance (GCV). In 2022WS, the GCV (17.71%) and PCV (23.46%) estimate for yield (GYTH) were medium to high. Heritability of this trait was high (56.96%), and GAPM (27.53%) was also high. In 2023DS, medium range of GCV (14.26%) and PCV (16.58%) were observed for this trait. The heritability (73.92%) and GAPM estimates were high (25.25%) for this trait. In 2022WS, the genotypic coefficient of variance (GCV) and phenotypic coefficient of variance (PCV) estimates for DFF were low (5.06%, 5.35%) and close to each other. Heritability estimates were high (89.58%) with low genetic advance as percent of mean (9.88%) values. In 2023DS, GCV and PCV values for DFF were low (3.61%, 4.14%). Broad sense heritability (76.32%) with low GAPM (6.50%) values were recorded for this trait.

## Correlation coefficient

The correlation coefficient evaluates how strongly two attributes are linked. In this study, a character association among reproductive traits revealed that yield and yield-related traits had both positive and negative relationships (Fig 3). In 2022WS, grain yield (GYTH) showed a significant positive correlation both at genotypic and phenotypic levels with DFF (0.3401***), DTM (0.36821***), PH (0.1766**), PL (0.20012**), and SF (0.18532**). On the contrary, a considerable nonsignificant positive correlation was observed with ETH (0.0268 ns), TGW (0.06137 ns) and LWR (0.0205 ns) both at genotypic and phenotypic levels. In 2023DS, GYTH showed significant positive correlation with DFF (0.2926***), DTM (0.2372***), PH (9.4349***), PL (0.345418***), TGW (0.2029**), SF (0.23132***). It showed negative significant relationship with ETH (−0.2029**) and nonsignificant negative correlation with LWR (−0.0695 ns) both at genotypic and phenotypic levels.

## Principal component analysis (PCA)

Principal Component Analysis (PCA) aids in the reduction of trait dimensionality, revealing key factors that collectively contribute to total variability. The study incorporated 216 rice genotypes for Principal Component Analysis (PCA) to analyze 8 traits related to earliness, yield, and yield attributing characteristics. The resulting Principal Components (PCs) revealed that only four PCs in 2022WS & three PCs in 2023DS exhibited eigenvalues surpassing 1.000, collectively explaining approximately 66.9% & 61.9% respectively of the total variability (Table 2). In 2022WS, PC1, with an eigenvalue of 1.85,

**Table 1. Variance components and heritability values of different traits of 2022WS & 2023DS.**

| Season | 2022WS | | | | | 2023DS | | | | |
|---|---|---|---|---|---|---|---|---|---|---|
| Traits | GCV | PCV | $h^2$ | GA | GAPM | GCV | PCV | $h^2$ | GA | GAPM |
| DFF | 5.06 | 5.35 | 89.58 | 8.64 | 9.87 | 3.61 | 4.13 | 76.32 | 7.77 | 6.50 |
| DTM | 3.62 | 3.87 | 87.56 | 8.30 | 6.98 | 2.49 | 2.68 | 86.47 | 7.21 | 4.77 |
| PH | 7.01 | 7.79 | 80.95 | 15.28 | 12.99 | 8.67 | 9.08 | 91.25 | 17.74 | 17.07 |
| ET | 9.13 | 13 | 49.28 | 1.18 | 13.20 | 10.28 | 13.57 | 57.48 | 1.89 | 16.06 |
| PL | 6.69 | 7.37 | 82.34 | 3.41 | 12.51 | 7.49 | 8.26 | 82.20 | 3.41 | 13.99 |
| SF | 10.50 | 12.42 | 71.53 | 13.51 | 18.30 | 10.82 | 12.09 | 80.10 | 15.42 | 19.95 |
| LWR | 5.52 | 5.58 | 97.84 | 0.61 | 11.26 | 4.82 | 5.22 | 85.25 | 0.47 | 9.18 |
| TGW | 7.61 | 8.34 | 83.37 | 0.35 | 14.32 | 9.11 | 9.95 | 83.82 | 0.40 | 17.18 |
| GYTH | 17.70 | 23.46 | 56.96 | 0.93 | 27.52 | 14.25 | 16.58 | 73.92 | 1.52 | 25.25 |

DFF- Days to 50% flowering; DTM- Days to Maturity; PH- Plant height (cm); ETH- Effective tillers per hill; PL- Panicle length (cm); SF- Spikelet fertility (%); TGW- Thousand grain weight; LWR- Grain length-width ratio, GYTH- Grain Yield (t/ha); GCV- Genotypic Coefficient of Variance; PCV- Phenotypic Coefficient of Variance; $h^2$- Heritability (Broad Sense); GA- Genetic Advance; GAPM- Genetic Advance as percentage of mean; WS- Wet Season; DS- Dry Season

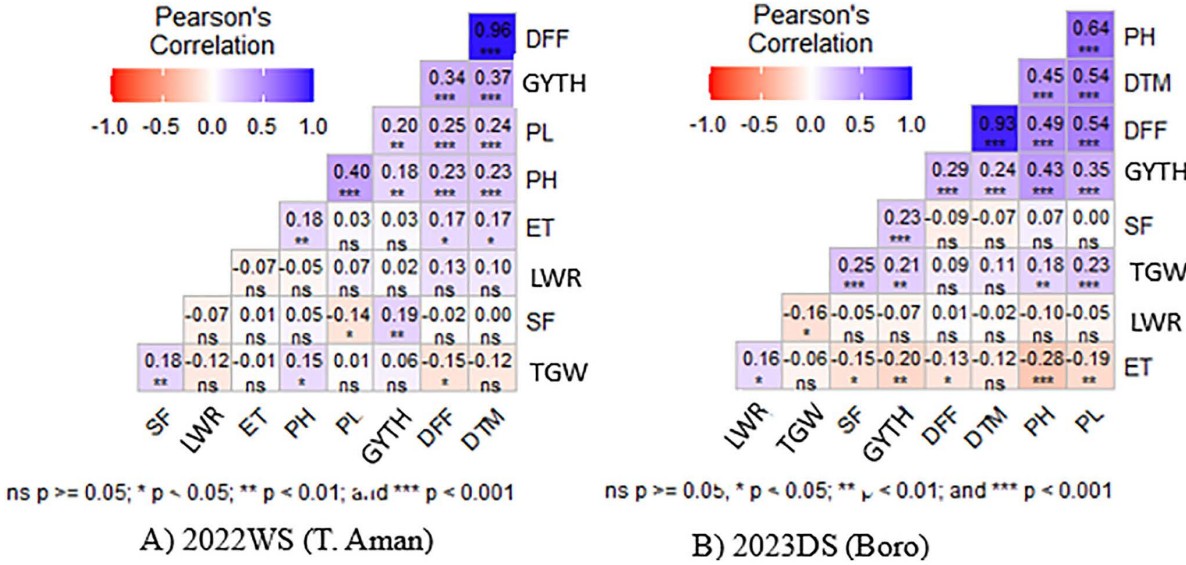

**Fig 3. Pearson correlation coefficient of nine quantitative traits in rice.**

**Table 2. Eigenvalues, % variance and cumulative eigenvalues of Rice genotypes.**

| | 2022WS | | | 2023DS | | |
|---|---|---|---|---|---|---|
| **PC** | **Eigenvalues** | **Variance (%)** | **Cum. variance (%)** | **Eigenvalues** | **Variance (%)** | **Cum. variance (%)** |
| PC1 | 1.85 | 23.10 | 23.10 | 2.57 | 32.10 | 32.10 |
| PC2 | 1.39 | 17.30 | 40.40 | 1.38 | 17.30 | 49.40 |
| PC3 | 1.10 | 13.80 | 54.20 | 1 | 12.50 | 61.90 |
| PC4 | 1.02 | 12.70 | 66.90 | 0.91 | 11.40 | 73.30 |
| PC5 | 0.87 | 10.80 | 77.80 | 0.68 | 8.55 | 81.80 |
| PC6 | 0.69 | 8.62 | 86.40 | 0.59 | 7.38 | 89.20 |
| PC7 | 0.59 | 7.35 | 93.70 | 0.53 | 6.62 | 95.80 |
| PC8 | 0.50 | 6.26 | 100 | 0.33 | 4.17 | 100 |

accounted for 23.1% of the variability, followed by PC2 (eigenvalue 1.39, 17.3% variability), PC3 (eigenvalue 1.1, 13.8% variability), and PC4 (eigenvalue 1.02, 12.7% variability) & in 2023DS, PC1, with an eigenvalue of 2.57, accounted for 32.1% of the variability, followed by PC2 (eigenvalue 1.38, 17.3% variability) and PC3 (eigenvalue 1, 12.5% variability). Subsequent PCs exhibited a gradual decline in variability.

## Genetic diversity analysis

K-means is the most popular clustering formulation in which the goal is to maximize the expected similarity between data items and their associated cluster centroids. K-means clustering was carried out on the evaluated parameters to assess the level of divergence among 216 rice genotypes. Using Elbow technique, Silhouette method, and Gap statistic method, which demonstrates that the plots display a bend in the graph is the appropriate number of clusters called k in k means algorithm. Elbow curve, i.e., within sum of squares, recommend employing 3 clusters for the final analysis and result extraction, which is the number of ideal clusters. The optimal number of k is 3, as shown by these charts in both seasons representing plots with 3 prominent bends (S1 Fig). Among the 3 clusters, cluster I accommodated the maximum number

of genotypes (97) in 2022WS, and cluster I accommodated the maximum number of genotypes (96) in 2023DS (Fig 4). The lowest numbers of genotypes were grouped in cluster III in both seasons.

The intra cluster distance of each cluster was zero (0.00), in both rice growing season indicates that the genotypes within the clusters are closely related or highly homogeneous in nature. The inter cluster distance in 2022WS ranged from 14.79 (cluster I and III) to 18.14 (cluster I and II). In 2023DS, the highest inter cluster distance was found between cluster II cluster III (21.40) and the lowest distance were between cluster I and II (14.99) (Table 3).

## Multi-Trait Genotype–Ideotype Distance Index (MGIDI) analysis

Multi-Trait Genotype Ideotype Distance Index (MGIDI) aims to capture significant genotype effects that are crucial for genetic factors in influencing the observed variation in traits. In 2022WS, analysis revealed distinct trait groupings considering four factors from the initial set of 8 traits, which collectively contribute to 66.90% of the total variation among traits. These factors provide a more streamlined perspective on the interrelatedness of traits. Factor 1 encompasses traits such as PH & PL, Factor 2 is associated with traits like DFF & TGW, Factor 3 is linked to traits including SF & GYTH and lastly, Factor 4 is associated with traits like ETH & LWR (Table 4). In case of 2023DS, three factors contributed for 61.90% of total variation. Factor 1 encompasses traits such as DFF, PH, PL & GYTH, Factor 2 is associated with traits like SF &

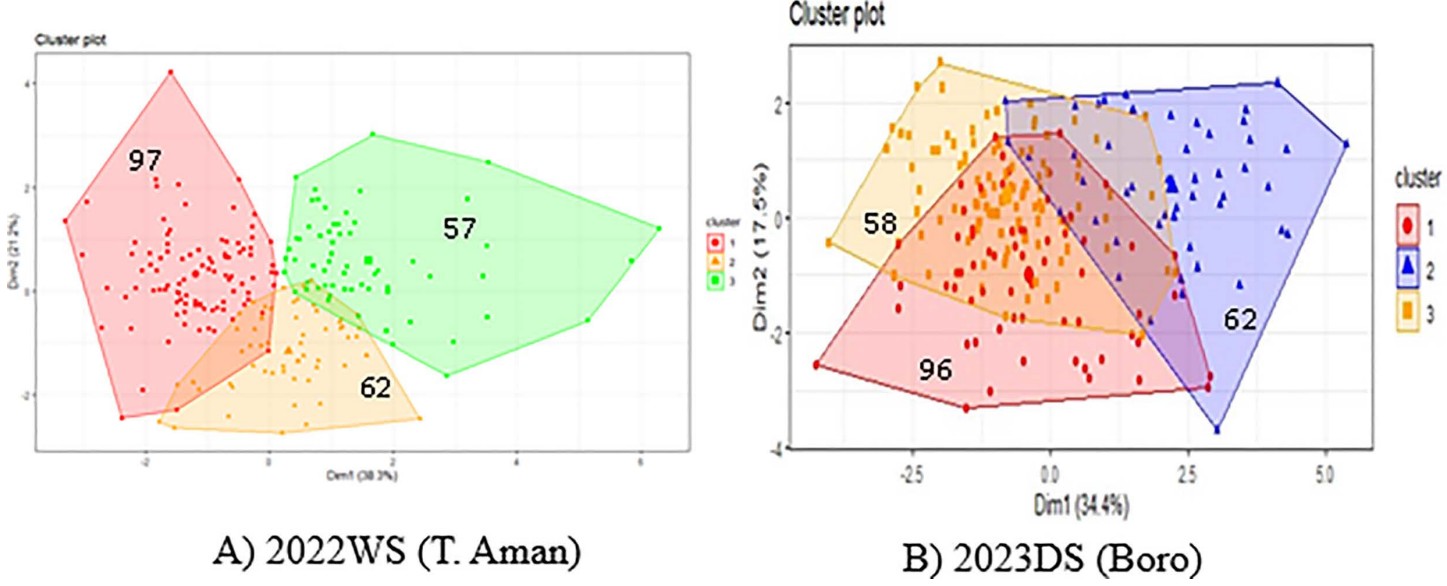

**Fig 4.  K-means clustering plot of A) 2022WS (T. Aman) and B) 2023DS (Boro).**

**Table 3.  Distances between cluster centroids based on K-means values of 216 rice genotypes for nine agronomic traits.**

|  |  | 2022WS |  |  |
|---|---|---|---|---|
|  |  | Cluster I | Cluster II | Cluster III |
| **2023DS** | Cluster I | **0.00** | 18.14 | 14.79 |
|  | Cluster II | 14.99 | **0.00** | 15.14 |
|  | Cluster III | 20.09 | 21.40 | **0.00** |

**Table 4. Factorial loadings, communalities, uniqueness, and selection gains (SG) based on the multi-trait genotype–ideotype distance index (MGIDI).**

**2022WS**

| VAR | FA1 | FA2 | FA3 | FA4 | Communality | Uniquenesses | SG (%) | Sense | Goal |
|-----|-----|-----|-----|-----|-------------|--------------|--------|-------|------|
| DFF | −0.39 | −0.58 | −0.40 | 0.13 | 0.67 | 0.33 | 2.66 | Increase | 100 |
| PH | −0.76 | 0.17 | −0.12 | 0.22 | 0.67 | 0.33 | 6.93 | Increase | 100 |
| ETH | −0.10 | −0.16 | −0.03 | 0.86 | 0.78 | 0.22 | 2.63 | Increase | 100 |
| PL | −0.84 | −0.09 | 0.04 | −0.11 | 0.72 | 0.28 | 5.75 | Increase | 100 |
| TGW | −0.21 | 0.76 | −0.18 | −0.05 | 0.66 | 0.34 | 0.81 | Increase | 100 |
| SF | 0.24 | 0.30 | −0.74 | 0.03 | 0.71 | 0.29 | 2.55 | Increase | 100 |
| LWR | −0.08 | −0.47 | −0.01 | −0.52 | 0.5 | 0.5 | 0.18 | Increase | 100 |
| GYTH | −0.26 | −0.14 | −0.75 | 0 | 0.65 | 0.35 | 7.01 | Increase | 100 |
| Average | | | | | **0.669** | | | | |

**2023DS**

| VAR | FA1 | FA2 | FA3 | Communality | Uniquenesses | SG (%) | Sense | Goal |
|-----|-----|-----|-----|-------------|--------------|--------|-------|------|
| DFF | −0.78 | −0.18 | 0 | 0.64 | 0.36 | 1.14 | Increase | 100 |
| PH | −0.8 | 0.18 | −0.18 | 0.7 | 0.3 | 2.26 | Increase | 100 |
| ETH | 0.25 | −0.11 | 0.63 | 0.47 | 0.53 | 3.42 | Increase | 100 |
| PL | −0.84 | 0.11 | −0.06 | 0.72 | 0.28 | 2.88 | Increase | 100 |
| TGW | −0.13 | 0.68 | −0.09 | 0.48 | 0.52 | 3.93 | Increase | 100 |
| SF | 0.12 | 0.81 | −0.05 | 0.68 | 0.32 | 3.99 | Increase | 100 |
| LWR | −0.07 | −0.04 | 0.86 | 0.75 | 0.25 | 4.94 | Increase | 100 |
| GYTH | −0.51 | 0.5 | −0.07 | 0.51 | 0.49 | 3.3 | Increase | 100 |
| Average | | | | **0.619** | | | | |

TGW and Factor 3 is linked to traits including ETH & LWR (Table 4). The average communality in 2022WS & 2023DS accounted for 66.9% and 61.9%, respectively, of all the genetic variability in the dataset, as outlined in Table 4.

In this study, 8 traits were employed to assess variations among the 216 rice genotypes in 2022WS & 2023DS season. Among the 216 rice genotypes evaluated, the Multi-Trait Genotype–Ideotype Distance Index (MGIDI) pinpointed 15% as selection intensity (SI) of the total material which is 32 accessions in both 2022WS & 2023DS season as high-performing for multiple traits, offering significant potential for the simultaneous improvement of 8 measured traits in rice breeding programs (Fig 5 and Table 5). The top 3 genotypes identified by the MGIDI index— IR19A8066 (3.25), IR19A7531 (3.68) & IR19A8052 (3.77) for 2022WS and IR19A8066 (5.3), IR19A7733 (5.69) & IR19A7430 (5.89) for 2023DS — emerge as promising candidates with exceptional characteristics (Table 5).

In Fig 5, the genotypes are arranged in descending order of MGIDI index values, with the highest value at the center and the lowest at the outer circle. The red circle represents the cut point according to the selection threshold (SI=15%) set by the MGIDI selection index. Genotypes were selected based on their MGIDI index, as indicated by red dots, while the unselected accessions are shown as black dots. IR19A8066 emerged as the most desirable genotype in both 2022WS & 2023DS according to the MGIDI index. Notably IR19A7833 & IR19A8520 respectively for 2022WS & 2023DS situated near the cut point indicated by the red line (Fig 5). The MGIDI index efficiently selected accessions IR19A8066, IR19A8052, IR19A7440, IR19A9061, and IR19A9054 as promising candidates as found in both 2022WS & 2023DS seasons top 15% genotypes (Table 5) for rice improvement programs.

The comprehensive assessment of strengths and weaknesses provided valuable insights, emphasizing the significance of an ideal rice genotype with improved quantitative traits. Fig 6 provides a comprehensive overview of the strengths and weaknesses exhibited by different genotypes, delineated by the contribution of each factor to the Multi-Trait Genotype–Ideotype Distance Index (MGIDI). The positioning of factors relative to genotypes indicates their influence, while dotted

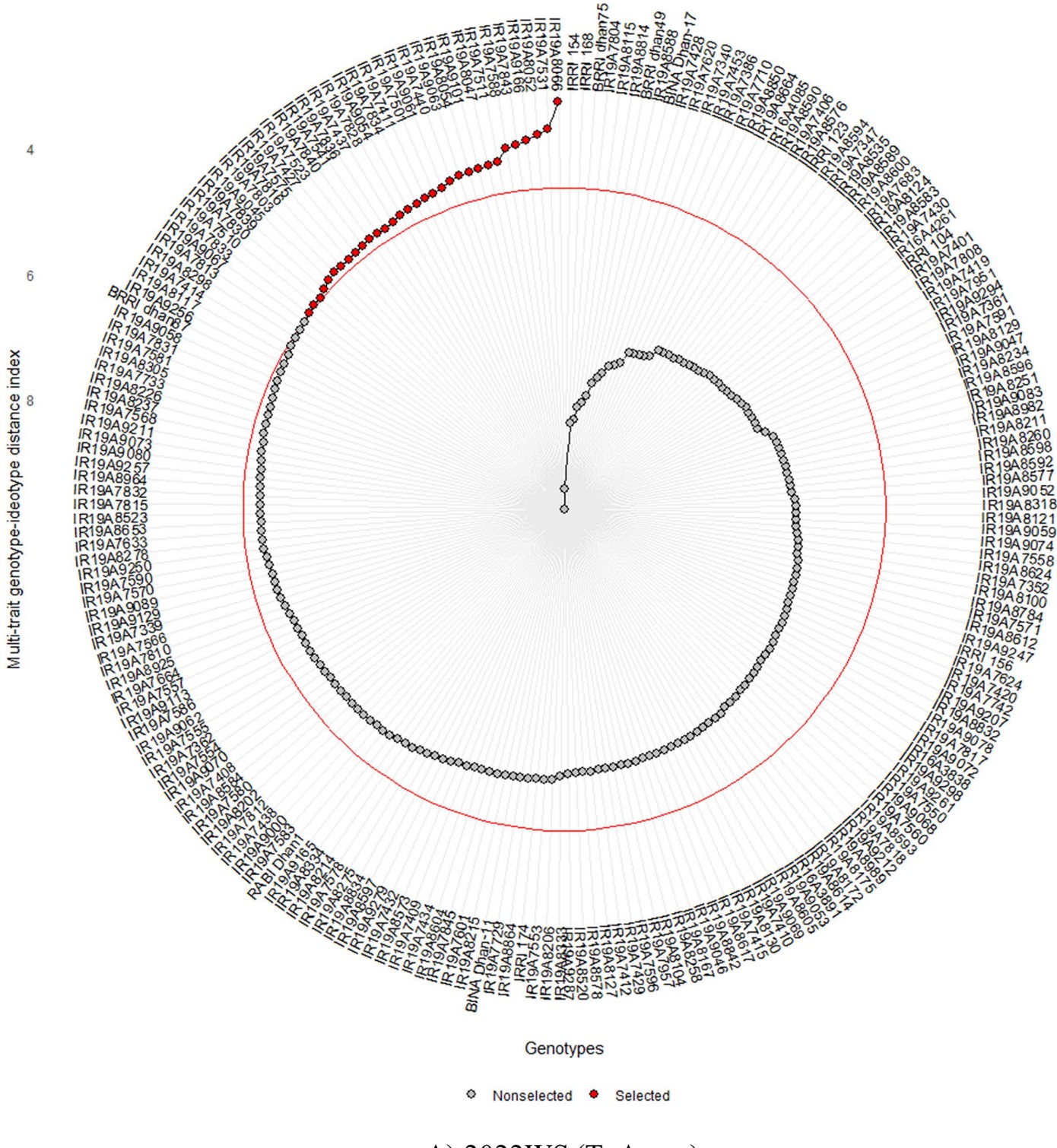

A) 2022WS (T. Aman)

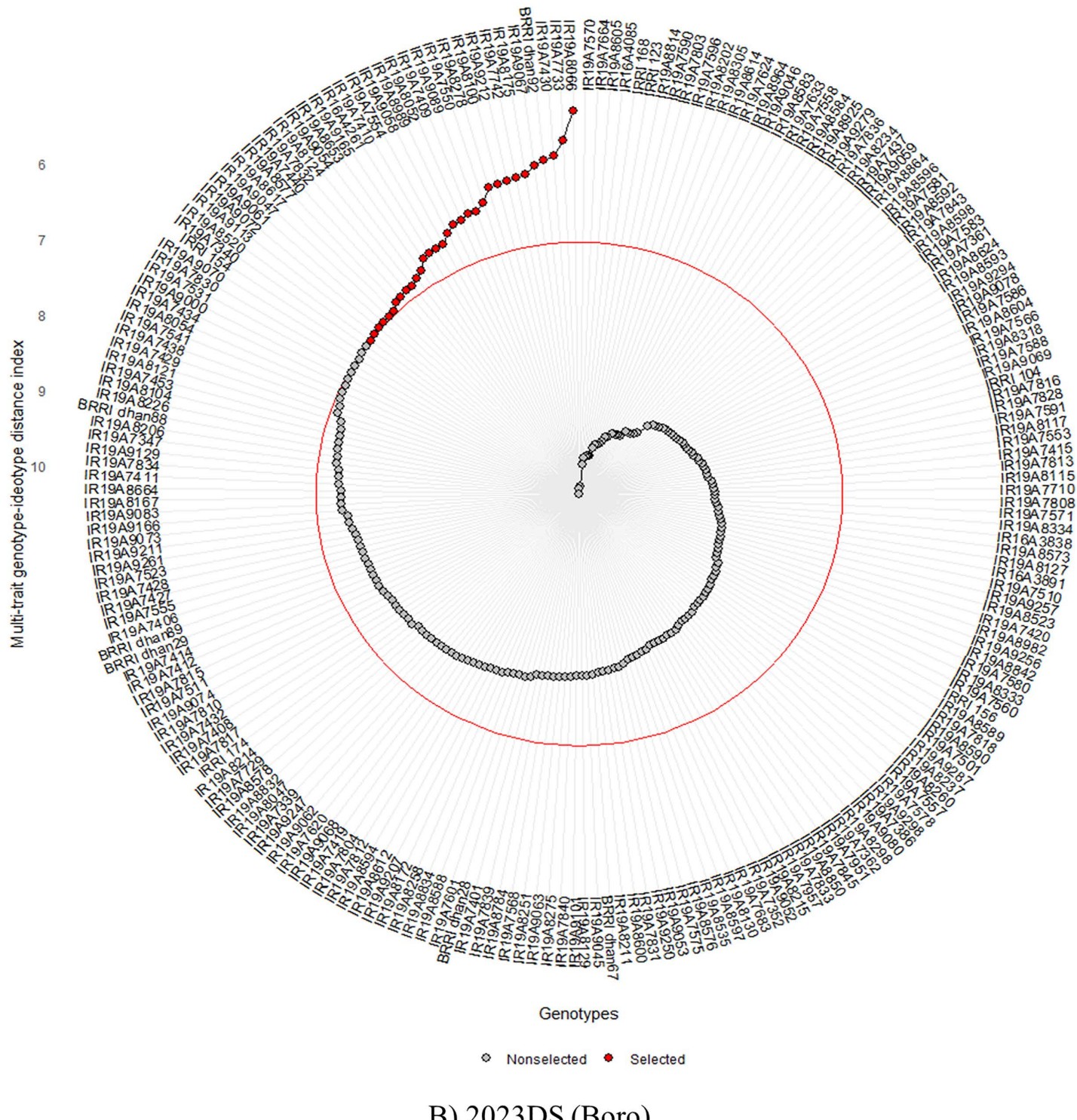

### B) 2023DS (Boro)

**Fig 5. Rice accession rankings showing selected accessions using the multi-trait genotype– ideotype index (MGIDI).**

**Table 5. The multi-trait genotype-ideotype distance index for Rice genotypes.**

| 2022WS | | | 2023DS | | |
|---|---|---|---|---|---|
| SN | Genotype | MGIDI | SN | Designation | MGIDI |
| 1 | **IR19A8066** | 3.25 | 1 | **IR19A8066** | 5.3 |
| 2 | IR19A7531 | 3.68 | 2 | IR19A7733 | 5.69 |
| 3 | **IR19A8052** | 3.77 | 3 | IR19A7430 | 5.89 |
| 4 | IR19A9166 | 3.84 | 4 | BRRI dhan92 | 5.92 |
| 5 | IR19A7843 | 3.89 | 5 | IR19A9067 | 5.99 |
| 6 | IR19A7588 | 3.93 | 6 | IR19A8175 | 6.09 |
| 7 | IR19A7511 | 4.11 | 7 | IR19A7742 | 6.11 |
| 8 | IR19A8047 | 4.13 | 8 | IR19A9212 | 6.12 |
| 9 | IR19A9101 | 4.16 | 9 | IR19A8100 | 6.14 |
| 10 | IR19A8054 | 4.17 | 10 | IR19A8278 | 6.15 |
| 11 | IR19A9063 | 4.17 | 11 | IR19A7550 | 6.33 |
| 12 | **IR19A7440** | 4.21 | 12 | IR19A9089 | 6.41 |
| 13 | **IR19A9061** | 4.27 | 13 | IR19A7409 | 6.41 |
| 14 | IR19A7501 | 4.3 | 14 | **IR19A8052** | 6.45 |
| 15 | IR19A7411 | 4.32 | 15 | IR19A8989 | 6.47 |
| 16 | IR19A7834 | 4.35 | 16 | IR19A9058 | 6.55 |
| 17 | **IR19A9054** | 4.36 | 17 | IR19A7554 | 6.65 |
| 18 | IR19A7828 | 4.38 | 18 | IR19A7410 | 6.65 |
| 19 | IR19A7437 | 4.42 | 19 | IR16A4261 | 6.66 |
| 20 | IR19A7836 | 4.44 | 20 | IR19A9165 | 6.69 |
| 21 | IR19A7541 | 4.44 | 21 | IR19A8653 | 6.8 |
| 22 | IR19A7840 | 4.45 | 22 | **IR19A9054** | 6.86 |
| 23 | IR19A7523 | 4.47 | 23 | IR19A8124 | 6.9 |
| 24 | IR19A7427 | 4.47 | 24 | IR19A7832 | 6.9 |
| 25 | IR19A7575 | 4.48 | 25 | **IR19A7440** | 6.92 |
| 26 | IR19A7816 | 4.49 | 26 | IR19A8577 | 6.94 |
| 27 | IR19A7803 | 4.49 | 27 | IR19A8617 | 6.99 |
| 28 | IR19A9045 | 4.5 | 28 | IR19A9047 | 7 |
| 29 | IR19A7839 | 4.55 | 29 | **IR19A9061** | 7 |
| 30 | IR19A7830 | 4.6 | 30 | IR19A9072 | 7.02 |
| 31 | IR19A7510 | 4.61 | 31 | IR19A9113 | 7.03 |
| 32 | IR19A7833 | 4.62 | 32 | IR19A8520 | 7.04 |

lines represent average performance in factor contribution. Higher factor values moving toward the center indicate weaknesses, while lower values signify strengths. In 2022WS, accessions associated with Factor 1 (FA1), such as IR19A7523, IR19A7531 and IR19A7541, demonstrate particular strengths in traits such as PH & PL (Fig 6 and Table 4). Conversely, accessions like IR19A9054, IR19A7440 and IR19A9061demonstrated strengths by contributing above average to FA2 linked with DFF & TGW, unlike IR19A8066 and IR19A8052, which exhibited below-average contribution, highlighting their weaknesses. In case of FA3 and FA4 all the genotypes showed above average contribution. Furthermore, in 2023DS season, genotypes IR19A8066, IR19A8052 and IR19A7440, associated with FA1, exhibit strength in traits like DFF, PH, PL & GYTH. Accessions like IR19A8278, IR19A8617 and IR19A9165, linked to FA2, showcase strength in traits like SF & TGW. Lastly, Factor 3 (FA3) with accessions like IR16A4261, IR19A7733, IR19A9067, and IR19A8175 demonstrates strength in traits like ETH & LWR (Fig 6 and Table 4).

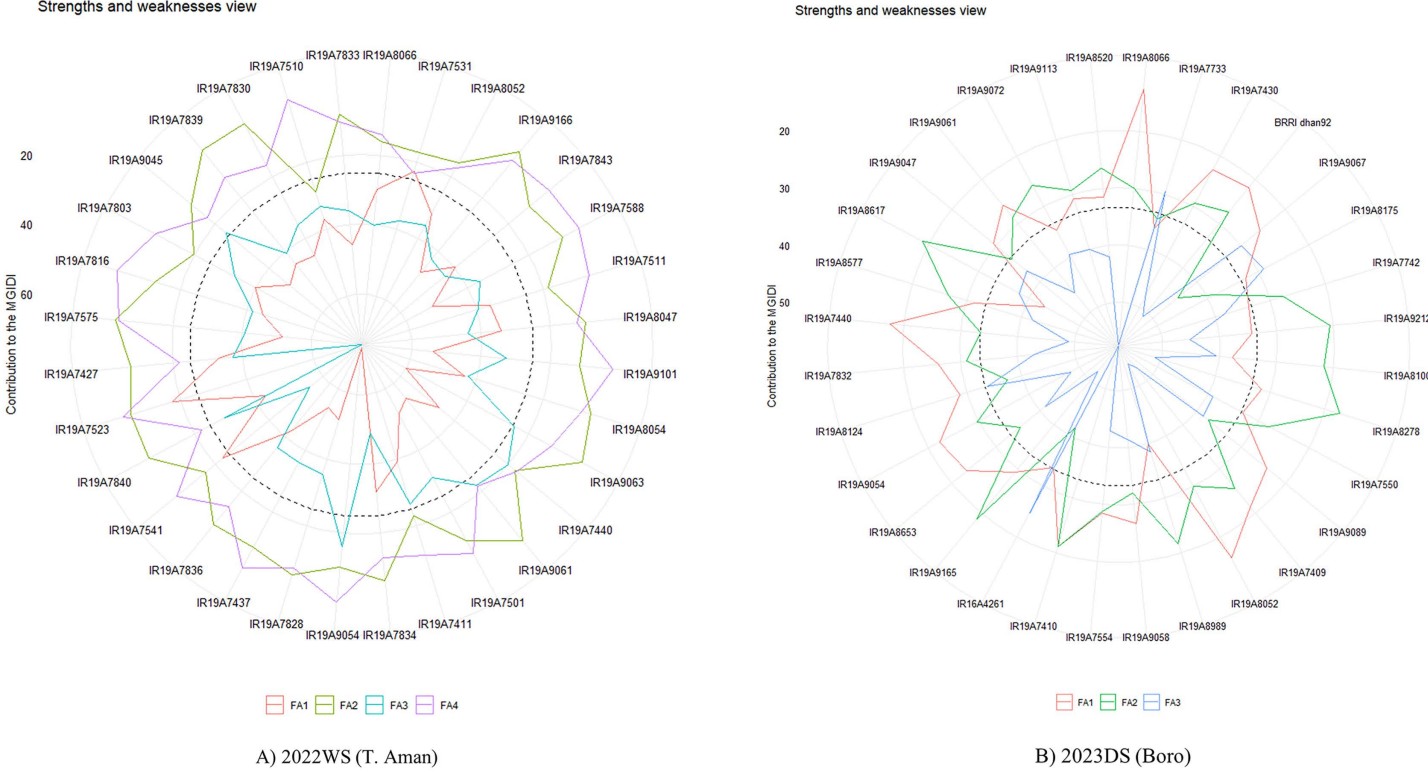

**Fig 6. The strengths and weaknesses of the selected genotypes are shown as the proportion of each factor on the computed multi-trait genotype-ideotype index (MGIDI).** The smaller the proportion explained by a factor (closer to the external edge), the closer the traits within that factor are to the ideotype. The black broken circle at the center shows the theoretical value if all the factors contributed equally.

## Breeding value estimation

Genomic estimated breeding values (GEBV) are based on additive effects and can help to identify suitable parental lines to cross for creating new breeding populations. Among 216 experimental materials excluding 16 check materials, all the studied genotypes had breeding values in a range of −0.55 to 0.68 (S3 Table). Among 200 genotypes, 100 genotypes possess positive GEBV, which ranged from 0 to 0.68 (Fig 7).

In order to identify the best breeding genotypes (hereafter called elite lines) the lines are filtered based on high ranking for their grain yield breeding values. The best performing lines in terms of breeding value for yield were selected and filtered based on the reliability of the breeding value estimate and their relatedness to other lines in the dataset based on pedigree. In this study, the reliability percentage of all the positive GEBVs lines was 58 to 85.9%. The highest yielder was IR19A8054 with a GEBV of 0.68, and the lowest yielder was IR19A9257 (0.0004). The genotypes having positive GEBVs (>0.5) are enlisted in Table 6 with reliability ranging 72–78%.

## Discussion

### Mean performance and ANOVA

The average performance showed a lot of variation among the advanced breeding lines. This implied the possibility of exploiting these genotypes in studying the inheritance of the traits and their inclusion in breeding programs for further improvement. The analysis of variance (ANOVA) revealed considerable variation across the tested genotypes for all the traits. Thus, there is ample scope for the selection of various quantitative traits for rice improvement. Significant genetic

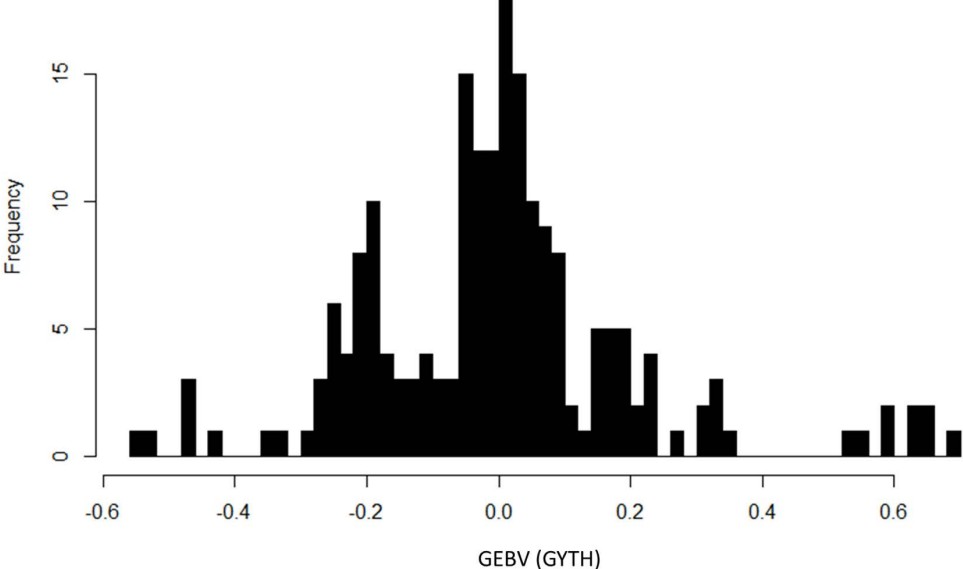

**Fig 7. Histogram showing the frequency distribution of 200 rice genotypes for Genomic estimated breeding values (GEBV) on yield.**

**Table 6. List of genotypes having positive GEBVs (>0.5).**

| Designation | GEBV (GYTH) | %-reliability |
|---|---|---|
| IR19A8054 | 0.68190403 | 0.75164394 |
| IR19A8052 | 0.64951309 | 0.76165449 |
| IR19A7501 | 0.64293523 | 0.74562708 |
| IR19A8047 | 0.63392122 | 0.75039232 |
| IR19A8066 | 0.62400483 | 0.74190894 |
| IR19A7531 | 0.59873685 | 0.77622036 |
| IR19A7523 | 0.5829385 | 0.78164561 |
| IR19A7541 | 0.55783956 | 0.77067423 |
| IR19A7510 | 0.53327839 | 0.72769561 |

differences among the rice genotypes were also reported by Barhate et al. (2021), Donkor et al. (2021), and Sarkar (2020) [58–61].

From the box plot of the OYT population, almost all studied trait boxes were narrow compared to other traits, indicating that the variation among the traits were lower. In both seasons, it was also noted that every character, aside from SF, was skewed towards receiving the lowest possible score. This indicated that most of the values fell in the minimum range. Additionally, it was noted that every character showed outliers; for example, DFF might be utilized for earliness, whereas GYTH was seen to be high yielding. Breeders can use these variations to obtain superior parental materials in several breeding cycle.

## Genetic variability

The current analysis shows that for most traits, the phenotypic coefficient of variance (PCV) was higher than but still very close to the corresponding genotypic coefficient of variance (GCV). PCV values were greater than their corresponding GCV values for the traits such as ETH, SF, and GYTH showing that environmental factors affect their expression. For the

traits PH, PL, SF, TGW, and GYTH, heritability was found to be high with a high value of GAPM. The results signify that additive gene action governs the expression of the trait PH, PL, SF, TGW, and GYTH. It will be worthwhile to make decisions based on these characteristics.

The GCV and PCV estimates revealed a lower range of variation present among the genotypes in respect of DFF. GCV and PCV values were close to each other which indicate negligible influence of environment on this trait. Barhate et al. (2021) [60] also reported low GCV and PCV values for this trait. High heritability with low GAPM indicated non-additive gene action and high heritability might be because of favorable environment. Selection according to this trait might be difficult or ineffective. Donkor et al. (2021) [61], and Sumnath et al. (2017) [62] reported high heritability along with low GAPM for this trait.

GCV and PCV value reveals a broader range of variability present among the genotype for GYTH, which is a prerequisite for varietal development through selection. High heritability coupled with high GAPM indicates a preponderance of additive genetic effect. So, direct selection based on this trait might be rewarded. Chakrabarty et al. (2019) [63] reported higher estimates of GCV, PCV, heritability, and GAPM values for this character.

## Correlation coefficient

In terms of correlation coefficient analysis, yield exhibited a positive significant correlation with DFF (0.3401***, 0.2926***), DTM (0.36821***, 0.2372***), PH (0.1766**, 0.4349***), PL (0.20012**, 0.345418***) and SF (0.185326**, 0.23132***) at genotypic and phenotypic levels in both season 2022WS and 2023DS. The findings suggested that the population's yield could be improved by screening genotypes with superior performance in the attributes DFF, DTM, PH, PL, and SPF.

Chakrabarty et al. (2019) [63] reported significant positive relationship with DTM and PL for DFF. Hossain et al. (2018) [64] reported DFF have significant negative association with panicle per hill (PPH) and GYTH. Kishore et al. (2018) [65] found DFF has significant positive correlation with DTM and PH and negative correlation with PPH. Result shows improvement of yield can be done by improving PH, SF and TGW. Hossain *et al.* (2018) [64] and Kishore *et al.* (2018) [65] also found grain yield is positively associated with PH, SPF and TGW.

## Principal component analysis (PCA)

Principal Component Analysis (PCA) aids in the reduction of trait dimensionality, revealing four and three key factors in 2022WS and 2023DS that collectively contribute to 66.9% & 61.9% of total variability with eigenvalues greater than one. The first principal component, representing a linear combination of the original predictor variables, captured the maximum variance in the dataset. Its selection is crucial, as it determines the direction of the highest variability. In this context, the substantial variability captured by PC1, with a 23.1% in 2022WS (eigenvalue 1.85) & 32.1% in 2023DS (eigenvalue 2.57) contribution, underscores its importance for guiding the selection of lines, making them particularly desirable for further breeding. Similar results were observed by Prasad et al. (2023) [66] & Pallavi et al. (2024) [24]. The remaining factors contribute minimally to the overall variability and are considered less critical. Consequently, only these four influential factors will be further considered in the calculation of the MGIDI index. This selective focus ensures a more targeted and meaningful analysis, aligning with the identified dimensions of maximum variance in the dataset.

## Genetic diversity analysis

Genetic diversity analysis based on K-means clustering grouped 216 genotypes into three clusters in both seasons. The genotypes located in one cluster differed from the other, whereas similarity prevailed within the cluster. The genotypes were in different aggregates based on dissimilarity. The genotypes of cluster I were more diverse than the genotypes of cluster III, whereas cluster I and cluster II had less diversity compared with the earlier one in both seasons. With a smaller inter-cluster distance between clusters, the genotypic diversity was lower. Similarly, the higher the inter-cluster distance,

the higher the genotypic diversity. Inter-cluster distances were in the range of 14.79 to 18.14 in 2022WS and 14.99 to 21.40 in 2023DS (Table 3), revealing variation between the clusters, which might be utilized through selection. Khatun *et al.* (2023) [67], Salunkhe et al. (2023) [68] also showed K-means clustering for measuring diversity.

## MGIDI

The Multi-trait Genotype–Ideotype Distance Index (MGIDI) highlight its significant potential as a robust and efficient multi-trait selection index, effectively integrating diverse phenotypic information in optimizing genetic gain to provide a clear and comprehensive ranking of genotypes. In 2022WS four factors were found with communality 0.669 and in 2023DS three factors with communality 0.619. This reaffirms the efficacy of factor analysis in establishing an index that optimally selects traits [26]. Evaluation of the MGIDI index demonstrated desired genetic gains across the analysed 8 traits, with a total genetic gain of 28.514% in 2022WS and 25.86% in 2023DS. Noteworthy traits such as GYTH, PL in 2022WS exhibited substantial percent selection gains of 7.01, 5.75 and in 2023DS, SF, TGW showed selection gains of 3.99, 3.93 respectively. This emphasizes the effectiveness of the MGIDI index in facilitating targeted and favourable trait selection for enhanced crop improvement strategies.

The versatility of the MGIDI model is underscored by its successful application in assessing ideal yield and yield-related traits in various crops such as Rice (Pallavi et al., 2024) [24], Maize (Palaniyappan et al., 2023) [69], Wheat (Meier et al., 2021) [70]. These diverse studies collectively highlight the efficiency of multivariate selection indices for simultaneous trait selection. Moreover, Olivoto and Nardino (2021) [26] have indicated that MGIDI stands out as the most efficient index for selecting genotypes with desired characteristics, further reinforcing its applicability and effectiveness. In this study, the MGIDI analysis identified 32 rice genotypes among 216 genotypes (SI = 15%) that consistently performed well across multiple desired traits in both the 2022WS and 2023DS. These varieties show great promise for simultaneously improving up to 8 different traits in rice breeding programs. Specifically, the top three genotypes, as determined by the MGIDI index, are particularly noteworthy. For the 2022WS, these include IR19A8066 (with an MGIDI score of 3.25), IR19A7531 (3.68), and IR19A8052 (3.77). In the 2023DS, the top performers were IR19A8066 (5.3), IR19A7733 (5.69), and IR19A7430 (5.89). These identified genotypes stand out as exceptional genotypes for further development due to their superiority. Additionally, from Fig 5, IR19A7833 & IR19A8520 found to be the last red dots, show interesting features that demand more research. Breeders are encouraged to closely examine these cut-point genotypes.

The visual representation of the analysis in Fig 6 illustrates each genotype's contribution to the MGIDI which offers crucial insights into identifying ideal rice genotypes by thoroughly assessing their strengths and weaknesses in various quantitative traits. It highlights how different factors contribute to the MGIDI, with lower factor values indicating strengths and higher values revealing weaknesses. During 2022WS, genotypes IR19A7523, IR19A7531, and IR19A7541 (associated with Factor 1) showed strong performance in PH and PL (Fig 6 and Table 4). Conversely, IR19A9054, IR19A7440, and IR19A9061 excelled above average in traits related to DFF and TGW for FA2, while IR19A8066 and IR19A8052 showed below-average weaker contribution in these areas. All genotypes performed above average for Factors 3 and 4. In case of 2023DS, IR19A8066, IR19A8052, and IR19A7440 (FA1) displayed strengths in DFF, PH, PL, and GYTH. IR19A8278, IR19A8617, and IR19A9165 (FA2) were strong in SF and TGW, and IR16A4261, IR19A7733, IR19A9067, and IR19A8175 (FA3) showed strength in ETH and LWR. These understandings of genotype strengths and weaknesses can be very helpful in guiding the choice of parents for next breeding initiatives. Jalalifar et al. (2023) [34], Mamun et al. (2022) [33] emphasized the importance of ideal rice genotypes found using MGIDI and their potential for better quantitative features as valuable resources for developing recombinant populations, aligning with sustainable and effective crop improvement strategies. The selected rice lines—including the ones this study focuses on—emerge as the best genotypes for upcoming rice breeding initiatives, significantly improving crop quality overall. This methodical use of factors and traits aids in the creation of robust and productive rice cultivars.

## Breeding value estimation

Predicting the offspring's breeding values with the aim of shortening breeding cycles was the foundation of the original GS concept [21]. Choosing and crossing parents based on breeding values for pertinent qualities is the most crucial choice a breeder takes [23]. Multi-season and multi-location evaluation is important because plant performance is strongly influenced by environmental factors and genotype x environment interaction. Multi-environment evaluation requires to improve reliability of breeding values (EBVs) by separating genetic effects from environmental noise. As a result, it becomes essential to thoroughly assess the program's current genotypes and choose a few high-performing lines to serve as the foundation for a gene pool. From there, selection for high breeding value can, in conjunction with other advancements, lead to increased rates of genetic gain [71]. Breeding cycles can be totally separated from the establishment of commercial lines by choosing parents solely based on GEBV, as shown by the creation and simulation of GS techniques in recent years [72,73].

Elite breeding lines are identified by means of stringent testing and selection procedures that have a high mean performance or breeding value for yield in addition to critical features for grain quality and yield protection. Breeding lines' merit as good parents is typically assessed using a metric known as "breeding value," which is determined by combining an individual's genotypic profile and phenotypic value. This is because both heritable and non-heritable variation can be found in the phenotypic data. This helps to explain why a breeding line with superior phenotypes isn't always regarded as appropriate parent material. Thus, it is important to employ breeding values to choose suitable parents for a breeding program [7]. IRRI's previous irrigation program, which had been in operation for 60 years, identified a base population based on breeding value using its database of multi-year advanced yield trial data [13]. Therefore, the secret to success in future recurrent selection breeding techniques and maximizing genetic gains is to identify the high-performing lines based on the breeding values for grain production or any major feature of interest that represents the whole variety of the entire breeding collection.

Breeders of rice must concentrate on population improvement breeding techniques that use elite lines as parents. In these methods, the parents of each breeding cycle are chosen based on high additive breeding values for grain production [74]. Genetic gain rates are higher when recurrent selection procedures are utilized to quickly recycle the best and high breeding value lines [75]. Estimated breeding values are incredibly helpful in determining which parents to choose to maximize genetic gain as the breeding value is reflected by the mean of the candidate's progeny when it is mated with random individuals [76]. Breeding value (i.e., the average performance of a given parent's offspring) estimates the additive worth of an individual by borrowing information from related lines in a phenotypic data set using pedigree or genome-wide marker data. A breeding value uses the relationship matrix to calculate a line's additive value, which is the main source of genetic variance passed on to its offspring, whereas a BLUP value for phenotypic performance takes into consideration both the additive and nonadditive genetic values of a line [75]. For making decisions about parental selection and assessing a line's relative "eliteness," this information is crucial. Genomic predictions can be used to select for more candidates in an earlier or better developed generation, boosting the pace of genetic gain [77]. To boost up genetic gain, genotypes with GEBVs above the average value should be chosen as parents in the next breeding cycle because each positive result implies that if these parents use a cyclic breeding program, their offspring will be better (μ+ GEBV) in the following generation. Top-performing genotypes based on high breeding values for grain yield were also recognized as a future elite breeding resource in addition to genetic gain estimations [16]. The genotypes having positive GEBVs (>0.5) can be used for this purpose (Table 6). Among 216 experimental materials excluding 16 check materials, the GEBVs for 100 genotypes were 0 to 0.68; for the other 100 genotypes, they were in the range of −0.55 to 0 (Fig 7). A positive GEBV suggests that an individual is predicted to have a genetic advantage for the trait of interest compared to the average of the population. The magnitude of positive GEBV can also provide a quantitative measure of the genetic advantage. Parents are selected largely for their breeding value for quantitative features in a completely modernized breeding program [75]. Khanna et al.

(2022,2024) [16,74] identified top-performing lines based on grain yield breeding values as an elite panel for implementing future population improvement-based breeding schemes. Juma *et al.* (2021) [13], Quddus *et al.* (2019) [78], Chen *et al.* (2018) also reported assessment of breeding value to create elite vital panel that reflects the genetic variety in the breeding programme with the highest heritable output.

## Conclusions

The identified genotypes, including IR19A8066, IR19A8052, IR19A7440, IR19A9061, and IR19A9054 in both 2022WS & 2023DS through MGIDI ranking underscores their potential for commercial release or utilization as key breeding materials for advancing rice breeding strategies. For parental selection in the hybridization programme, IR19A8054, IR19A8052, IR19A7501, IR19A8047, IR19A8066, IR19A7531, IR19A7523, IR19A7541, and IR19A7510 with high positive GEBV (0.5) for yield are preferred. Genotype IR19A8784 was discovered two days ahead of check varieties BRRI dhan75 (109 days in 2022WS) and BRRI dhan67 (149 days in 2023DS) in both rice growing seasons. In 2023DS, IR19A8066 produced 8.11 t/ha, which is higher than the check variety BRRI dhan89 (7.97 t/ha), and IR19A9212 (7.12 t/ha) and IR19A9054 (6.85 t/ha) yielded better than check BRRI dhan28 and BRRI dhan67 (6.69 t/ha). Genotypes IR19A8066, IR19A9212, and IR19A9054 outperformed the check BINA dhan1 (3.99 t/ha) with 4.24, 4.08, and 4.81 t/ha in 2022WS. In addition, the positive GEBVs of IR19A8066, IR19A9212, and IR19A9054 were 0.624, 0.323, and 0.193, respectively. The aforementioned lines need to be reevaluated and redefined through more research before being released in desirable varieties.

## Supporting information

**S1 Fig. Elbow curve of K-means cluster.**
(DOCX)

**S1 Table. List of Genotypes.**
(DOCX)

**S2 Table. Analysis of variance (ANOVA) of nine morphological traits & mean performance of the breeding lines in 2022WS and 2023DS.**
(DOCX)

**S3 Table. List of Genomic estimated breeding values (GEBV).**
(DOCX)

## Acknowledgments

The Gazipur Agricultural University, Gazipur 1706, Bangladesh, has provided logistical support, which the author(s) gratefully recognized. The author(s) also acknowledge the support of the International Rice Research Institute and the Bill and Melinda Gates Foundation during implementing the project named "Accelerating the Genetic Gains in Rice (AGGRi): IRRI-NARES breeding networks using rapid-cycle genomic selection to deliver annual genetic gains of 2% in rice (C-2019-70)".

## Author contributions

**Conceptualization:** S. S. Chaity, M. R. Islam, J. U. Ahmed, A. K. M. Aminul Islam.

**Data curation:** S. S. Chaity, M. R. Islam, M. Faruquee, A. K. M. Aminul Islam.

**Formal analysis:** S. S. Chaity, M. Faruquee.

**Funding acquisition:** A. K. M. Aminul Islam.

**Investigation:** S. S. Chaity.

**Methodology:** S. S. Chaity, J. U. Ahmed, A. K. M. Aminul Islam.

**Project administration:** A. K. M. Aminul Islam.

**Resources:** A. K. M. Aminul Islam.

**Software:** M. R. Islam, M. Faruquee.

**Supervision:** M. R. Islam, J. U. Ahmed, A. K. M. Aminul Islam.

**Validation:** S. S. Chaity, M. Faruquee.

**Visualization:** S. S. Chaity, M. Faruquee.

**Writing – original draft:** S. S. Chaity.

**Writing – review & editing:** M. R. Islam, J. U. Ahmed, A. K. M. Aminul Islam.

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
