## [Decision Letter · Decision Letter 0]

22 Oct 2025

Dear Dr. Islam,

Thank you for submitting your manuscript to PLOS ONE. After careful consideration, we feel that it has merit but does not fully meet PLOS ONE’s publication criteria as it currently stands. Therefore, we invite you to submit a revised version of the manuscript that addresses the points raised during the review process.

**ACADEMIC EDITOR:**

The manuscript has been reviewed by three expert reviewers. Based on their evaluations, we believe the manuscript may be suitable for publication **pending major revisions** .

We kindly request that you carefully revise the manuscript in accordance with the reviewers' comments and suggestions. Please submit the revised version along with a detailed point-by-point response outlining how each comment has been addressed.

We look forward to receiving your revised submission.

Thanks

We look forward to receiving your revised manuscript.

Kind regards,

C Anilkumar, Ph.D.

Academic Editor

PLOS ONE

https://doi.org/10.1186/s12284-021-00533-5

https://doi.org/10.1016/j.dib.2024.110176

10.9734/CJAST/2022/v41i333950

In your revision ensure you cite all your sources (including your own works), and quote or rephrase any duplicated text outside the methods section. Further consideration is dependent on these concerns being addressed.

Additional Editor Comments:

The manuscript has been reviewed by three expert reviewers. Based on their evaluations, we believe the manuscript may be suitable for publication pending major revisions.

We kindly request that you carefully revise the manuscript in accordance with the reviewers' comments and suggestions. Please submit the revised version along with a detailed point-by-point response outlining how each comment has been addressed.

We look forward to receiving your revised submission.

Thanks

Reviewers' comments:

Reviewer's Responses to Questions

**Comments to the Author**

1. Is the manuscript technically sound, and do the data support the conclusions?

Reviewer #1: Yes

Reviewer #2: Yes

Reviewer #3: Yes

2. Has the statistical analysis been performed appropriately and rigorously?

Reviewer #1: Yes

Reviewer #2: Yes

Reviewer #3: Yes

3. Have the authors made all data underlying the findings in their manuscript fully available?

Reviewer #1: Yes

Reviewer #2: Yes

Reviewer #3: Yes

4. Is the manuscript presented in an intelligible fashion and written in standard English?

Reviewer #1: Yes

Reviewer #2: Yes

Reviewer #3: Yes

Reviewer #1: The manuscript which looked at identifying of elite rice Lines with better breeding values using genomic prediction and multi-trait Genotype Ideotype Distance Index (MGIDI) for grain yield under irrigation cropping system was well written with good statistical analysis and presentation. However, the tile can be modified to identification of elite rice Lines with better breeding values using genomic prediction and multi-trait Genotype Ideotype Distance Index (MGIDI) for grain yield under irrigation cropping system instead of eco system, because eco system means more than just a field with high level of moisture.

Reviewer #2: Reviewers’ Comments

The manuscript entitled “Identification of Elite Lines of Rice with Better Breeding Values using Genomic Prediction and Multi-trait Genotype Ideotype Distance Index (MGIDI) for grain yield under Irrigated Eco-System” presents breeding strategies to improve rice production under irrigated eco-system. The article is interesting and potentially impactful for researchers worldwide. The content is appropriate, and the authors explain the sections clearly. The study is suitable for publication after the following revisions:

1. In abstract what is DTM? And mention location where this trial was conducted.

2. Use only two data points after decimal in whole MS.

3. All abbreviations explain at least first time like BBS in introduction, follow same pattern in whole MS. Use consistence same abbreviations as GYTH in abstract but fig. 2 used YTH, crossed check all abbreviations in table, figures and MS.

4. For genomic selection use “GS” in whole MS.

5. Many of places have “percent” replace with “%”

6. In introduction mention some earlier study where they use “MGIDI” for genetic gain in other important crops.

7. In material and method: which generation of 200 advanced breeding lines?

8. Need discussion section to improve many sentences of overlapping with results, might be author just give indication of table or figure instead to giving all information again.

9. Discussion section includes why multi season/location is important for estimating breeding value and GS.

10. Overall genotyping related information very less in MS, it should be more emphasis.

11. References are not formatted according to journal special in discussion section.

Reviewer #3: The paper “Identification of Elite Lines of Rice with Better Breeding Values using Genomic Prediction and Multi-trait Genotype Ideotype Distance Index (MGIDI) for grain yield under Irrigated Eco-System‘’ aimed to select elite parental materials and this study evaluated 200 International Rice Research Institute (IRRI) developed advanced breeding lines during 2022 Wet Season (WS-T. Aman) and 2023 Dry Season (DS-Boro) along with two sets of check (10 global and 6 different local for each season) using an alpha lattice design with two replications

The paper is prepared professionally. It includes a well-crafted abstract and an exhaustive introduction that justifies the research undertaken. The introduction points to the deficiencies in the literature on the subject. The aim is clearly defined. Modern analytical methods were used in the research. The discussion of the results is well prepared. The conclusions are well-defined. The illustrative material is appropriate.

In my opinion, the manuscript after minor corrections, will be suitable for publication in a journal.

Detailed comments:

Abstract: Should include some more numeric data obtained from the study

Do not use abbreviations when use first time.

Introduction - The introduction is enough in my opinion. Introduction needs some minor changes

Rice is one of the major cereal grains farmed as a staple diet for more than half of the world's

population. It is a semi-aquatic grass plant that belongs to the Gramineae (Poaceae) family's

genus Oryza. This sentence needs references. I can suggest below ones

Zhang L, Suo L, Shan Y, Wang B (2024). Glutathione alleviates boron toxicity by modulating nitrogen metabolism and boronuptake in rice seedlings. Turkish Journal of Agriculture and Forestry 48 (6): 1011-1022. https://doi.org/10.55730/1300-011X.3237.

Habib, M.A., Azam, M.G., Haque, M.A. et al. Climate-smart rice (Oryza sativa L.) genotypes identification using stability analysis, multi-trait selection index, and genotype-environment interaction at different irrigation regimes with adaptation to universal warming. Sci Rep 14, 13836 (2024). https://doi.org/10.1038/s41598-024-64808-9

The purpose of plant breeding is to assemble more desirable combinations of genes or traits in

new varieties, and parents should have performed as magnificent contributors to one or more

traits that are being targeted in the breeding program [11]. Needs more references. I can suggest below ones

Zafar MM, Razzaq A, Anwar Z, Ijaz A, Zahid M, Iqbal MS, Farid G, Seleiman MF, Zaman RQ, Rauf A, Jiang X(2025). Enhancing salt tolerance and yield potential in cotton: insights from physiological responses, genetic variability, and heterosis. Turkish Journal of Agriculture and Forestry 49 (1): 110-124. https://doi.org/10.55730/1300-011X.3252

Any coordinate of experimental area?????

Tables should be arranged better way?

**Do you want your identity to be public for this peer review?** For information about this choice, including consent withdrawal, please see our Privacy Policy

Reviewer #1: **Yes:** Esther Fobi Donkor(PhD)

Reviewer #2: **Yes:** Gurjeet Singh

Reviewer #3: No

---

## [Author Response · Author response to Decision Letter 1]

25 Nov 2025

Dear Editor

We have made revisions of our manuscript as per the comments and suggestions of the reviewer in track change mode and highlighted by red font color. Detail response to reviewer comments are given below for your necessary information and action.

Regards

Corresponding Author

Academic editor:

Reply: Manuscript main body template is followed. Author name we provided can be published as it is.

https://doi.org/10.1186/s12284-021-00533-5 (Juma et al 2021)

https://doi.org/10.1016/j.dib.2024.110176

10.9734/CJAST/2022/v41i333950

In your revision ensure you cite all your sources (including your own works), and quote or rephrase any duplicated text outside the methods section. Further consideration is dependent on these concerns being addressed.

Reply: All 3 citation is included in the revised manuscript.

Reply: Data sets generated during experimentation and analysis of the current research are available from the corresponding author upon reasonable request.

Reply: All the figures in .tif file format has been saved following journal rules.

Reply: Reviewer’s recommended citation has been included in the revised manuscript.

Review Comments to the Author:

Reviewer #1: The manuscript which looked at identifying of elite rice Lines with better breeding values using genomic prediction and multi-trait Genotype Ideotype Distance Index (MGIDI) for grain yield under irrigation cropping system was well written with good statistical analysis and presentation. However, the tile can be modified to identification of elite rice Lines with better breeding values using genomic prediction and multi-trait Genotype Ideotype Distance Index (MGIDI) for grain yield under irrigation cropping system instead of eco system, because eco system means more than just a field with high level of moisture.

Reply: I have changed the title following the suggested one at line no. 1-3.

Reviewer #2: Reviewers’ Comments

The manuscript entitled “Identification of Elite Lines of Rice with Better Breeding Values using Genomic Prediction and Multi-trait Genotype Ideotype Distance Index (MGIDI) for grain yield under Irrigated Eco-System” presents breeding strategies to improve rice production under irrigated eco-system. The article is interesting and potentially impactful for researchers worldwide. The content is appropriate, and the authors explain the sections clearly. The study is suitable for publication after the following revisions:

1. In abstract what is DTM? And mention location where this trial was conducted.

Reply: I have added the meaning of DTM as Days to maturity (DTM) in the abstract section at line no. 23. Location of the experiment is also added at line no. 17-18.

2. Use only two data points after decimal in whole MS.

Reply: Only two data points after decimal is kept in the whole manuscript.

3. All abbreviations explain at least first time like BBS in introduction, follow same pattern in whole MS. Use consistence same abbreviations as GYTH in abstract but fig. 2 used YTH, crossed check all abbreviations in table, figures and MS.

Reply: In whole manuscript all abbreviations are explained at first time eg. For grain yield (ton/ha) – GYTH is used replacing YTH in whole manuscript, figure, table as well as abstract. at the line no. 248, 255, 268, Table 1.

4. For genomic selection use “GS” in whole MS.

Reply: GS is used instead of genomic selection following comment 4 of reviewer2 at the line no. 89, 223, 229, 550, 557.

5. Many of places have “percent” replace with “%”

Reply: Percent is replaced by % following comment 5 of reviewer2 at the line no. 56, 95

6. In introduction mention some earlier study where they use “MGIDI” for genetic gain in other important crops.

Reply: Some earlier study of other important crops is mentioned in introduction part at line no. 117-119 where MGIDI has been used.

7. In material and method: which generation of 200 advanced breeding lines?

Reply: The generation of 200 advanced breeding lines as F7 generation is added in material and method section at line no. 129.

8. Need discussion section to improve many sentences of overlapping with results, might be author just give indication of table or figure instead to giving all information again.

Reply: Discussion section has been thoroughly rechecked to avoid overlapping information with result section.

9. Discussion section includes why multi season/location is important for estimating breeding value and GS.

Reply: Added, Multi season and multi-location evaluation is important because plant performance is strongly influenced by environmental factors and interaction with the genotypes. It requires to improve reliability of breeding values (EBVs) by separating genetic effects from environmental noise.

10. Overall genotyping related information very less in MS, it should be more emphasis.

Reply: We did the field phenotyping and multi environment evaluation of IRRI developed advanced lines. Genotypic data we shared from IRRI, only to estimate breeding value.

11. References are not formatted according to journal special in discussion section.

Reply: References format has been changed to PLOS ONE journal format.

Reviewer #3: The paper “Identification of Elite Lines of Rice with Better Breeding Values using Genomic Prediction and Multi-trait Genotype Ideotype Distance Index (MGIDI) for grain yield under Irrigated Eco-System‘’ aimed to select elite parental materials and this study evaluated 200 International Rice Research Institute (IRRI) developed advanced breeding lines during 2022 Wet Season (WS-T. Aman) and 2023 Dry Season (DS-Boro) along with two sets of check (10 global and 6 different local for each season) using an alpha lattice design with two replications

The paper is prepared professionally. It includes a well-crafted abstract and an exhaustive introduction that justifies the research undertaken. The introduction points to the deficiencies in the literature on the subject. The aim is clearly defined. Modern analytical methods were used in the research. The discussion of the results is well prepared. The conclusions are well-defined. The illustrative material is appropriate.

In my opinion, the manuscript after minor corrections, will be suitable for publication in a journal.

Detailed comments:

Abstract: Should include some more numeric data obtained from the study

Do not use abbreviations when use first time.

Reply: Some numeric data of correlation obtained from the study has been included in the abstract part at line no. 22-24. Abbreviations at first time using has also been avoided in this section.

Introduction - The introduction is enough in my opinion. Introduction needs some minor changes

Rice is one of the major cereal grains farmed as a staple diet for more than half of the world's

population. It is a semi-aquatic grass plant that belongs to the Gramineae (Poaceae) family's

genus Oryza. This sentence needs references. I can suggest below ones

Zhang L, Suo L, Shan Y, Wang B (2024). Glutathione alleviates boron toxicity by modulating nitrogen metabolism and boronuptake in rice seedlings. Turkish Journal of Agriculture and Forestry 48 (6): 1011-1022. https://doi.org/10.55730/1300-011X.3237.

Habib, M.A., Azam, M.G., Haque, M.A. et al. Climate-smart rice (Oryza sativa L.) genotypes identification using stability analysis, multi-trait selection index, and genotype-environment interaction at different irrigation regimes with adaptation to universal warming. Sci Rep 14, 13836 (2024). https://doi.org/10.1038/s41598-024-64808-9

Reply: Both the references have been included in the introduction part at line no. 44.

The purpose of plant breeding is to assemble more desirable combinations of genes or traits in

new varieties, and parents should have performed as magnificent contributors to one or more

traits that are being targeted in the breeding program [11]. Needs more references.

I can suggest below ones

Zafar MM, Razzaq A, Anwar Z, Ijaz A, Zahid M, Iqbal MS, Farid G, Seleiman MF, Zaman RQ, Rauf A, Jiang X(2025). Enhancing salt tolerance and yield potential in cotton: insights from physiological responses, genetic variability, and heterosis. Turkish Journal of Agriculture and Forestry 49 (1): 110-124. https://doi.org/10.55730/1300-011X.3252

Reply: This reference has also been included in the introduction part at line no. 69.

Any coordinate of experimental area?????

Reply: Location of the experiment is added in the abstract at line no. 17-18 and in materials and method section at the line no. 137-138.

---

## [Decision Letter · Decision Letter 1]

17 Dec 2025

Identification of Elite Rice Lines with Better Breeding Values using Genomic Prediction and Multi-trait Genotype Ideotype Distance Index (MGIDI) for grain yield under Irrigation Cropping System

PONE-D-25-47900R1

Dear Dr. Islam,

We’re pleased to inform you that your manuscript has been judged scientifically suitable for publication and will be formally accepted for publication once it meets all outstanding technical requirements.

Kind regards,

C Anilkumar, Ph.D.

Academic Editor

PLOS One

Additional Editor Comments (optional):

Dear authors,

Thanks for making revisions in the manuscript. I and all three expert reviewers are happy about the revisions made, which improved the manuscript significantly. I recommend for the acceptance of the manuscript for publication.

Reviewers' comments:

Reviewer's Responses to Questions

**Comments to the Author**

Reviewer #1: All comments have been addressed

Reviewer #2: All comments have been addressed

Reviewer #3: All comments have been addressed

2. Is the manuscript technically sound, and do the data support the conclusions?

Reviewer #1: Yes

Reviewer #2: Yes

Reviewer #3: Yes

3. Has the statistical analysis been performed appropriately and rigorously?

Reviewer #1: Yes

Reviewer #2: Yes

Reviewer #3: Yes

4. Have the authors made all data underlying the findings in their manuscript fully available?

Reviewer #1: Yes

Reviewer #2: Yes

Reviewer #3: Yes

5. Is the manuscript presented in an intelligible fashion and written in standard English?

Reviewer #1: Yes

Reviewer #2: Yes

Reviewer #3: Yes

Reviewer #1: The authors have addressed the concerns of the reviewers so the manuscript can be accepted for publication

Reviewer #2: Dear Author,

The authors have carefully addressed all the comments and have no further suggestions.

Reviewer #3: Dear Editor,

The author (s) made all necessary changes and corrections on revised version. I believe that the paper is now ready for publication.

**Do you want your identity to be public for this peer review?** For information about this choice, including consent withdrawal, please see our Privacy Policy

Reviewer #1: **Yes:** Esther Fobi Donkor

Reviewer #2: **Yes:** Gurjeet Singh

Reviewer #3: No

---

## [Editor Report · Acceptance letter]

PONE-D-25-47900R1

PLOS One

Dear Dr. Islam,

I'm pleased to inform you that your manuscript has been deemed suitable for publication in PLOS One. Congratulations! Your manuscript is now being handed over to our production team.

Kind regards,

on behalf of

Dr. C Anilkumar

Academic Editor

PLOS One